

# Spillover of a biological control agent (*Chrysolina quadrigemina*) onto native St. Johnswort (*Hypericum punctatum*)

Jessica L. Tingle[1], Susan C. Cook-Patton[2] and Anurag A. Agrawal[3]

[1] Department of Biology, University of California, Riverside, Riverside, CA, United States
[2] Science & Technology Policy Fellow, American Association for the Advancement of Science, Washington, D.C., United States
[3] Department of Ecology and Evolutionary Biology, Cornell University, Ithaca, NY, United States

## ABSTRACT

Biological control agents may have unintended effects on native biota, particularly species that are closely related to the target invader. Here, we explored how *Chrysolina quadrigemina*, a beetle introduced to control the invasive weed *Hypericum perforatum*, impacts native *H. punctatum* in Tompkins County, New York, USA. Using a suite of complementary field surveys and experimental manipulations, we examined beetle preference for native and exotic *Hypericum* species and whether beetle herbivory influences the spatial distribution of *H. punctatum*. We found that the introduced beetle readily consumes native *H. punctatum* in addition to its intended target, and that *H. punctatum* at our field sites generally occurs along forest edges despite higher performance of experimental plants in more open habitats. However, we found no evidence that the beetle limits *H. punctatum* to forest edge habitats.

## INTRODUCTION

Invasions by exotic species can negatively impact native communities through altered species interactions and ecosystem processes (*Mack et al., 2000*). One explanation for the spread of invasive species is the *enemy release hypothesis*, which posits that invaders have higher performance than natives do because they escape from specialist enemies like herbivores or pathogens when they move to a new range (*Keane & Crawley, 2002*; *Vilà, Maron & Marco, 2005*; *Agrawal et al., 2005*). Thus, a logical way to combat invasive species is through biological control, the practice of releasing specialist enemies, typically from the original range of the invader, to control the spread of the exotic species (*McFadyen, 1998*; *Van Driesche et al., 2010*).

Ideal biological control agents are those that specialize on the target invader and do not spill over onto non-target, native hosts (*Chalak et al., 2010*). However, because the traits mediating plant-herbivore interactions, such as plant chemical defenses, are often phylogenetically conserved (*Futuyma & Agrawal, 2009*), most herbivores readily utilize several closely related plants (*Jaenike, 1990*; *Pearse & Altermatt, 2013*). This tendency poses

Corresponding author
Jessica L. Tingle,
jessica.tingle@email.ucr.edu

a challenge for the selection of biological control agents because specialist insects that are released to control invasive plant species may exploit closely related natives, particularly if those natives co-occur with the invasive plant (*Louda et al., 2003*).

The exotic beetle *Chrysolina quadrigemina* has successfully controlled exotic St. Johnswort, *Hypericum perforatum*, in California (*Van Driesche et al., 2010*), but in the eastern United States, it may have unintended and negative effects on a congeneric and co-occurring native, *H. punctatum*. The exotic plant is a Eurasian species that became a pest after its introduction to North America (*Bourke, 2000*). The beetle was introduced in 1943 as an attempt to control the weed, and is a classic case of successful biological control (*Huffaker, 1959*; *Van Driesche et al., 2010*). In addition to directly consuming the plant, the beetle also acts as a vector for the pathenogenic fungus *Colletotrichum gloeosporioides*, which may additionally help to control the exotic *H. perforatum* (*Morrison, Reekie & Jensen, 1998*).

Our study was motivated by an observation of *C. quadrigemina* feeding on the native *H. punctatum* in the field (Fig. 1). We formed three hypotheses. First, that *C. quadrigemina* readily consumes the native *H. punctatum*. Second, that *C. quadrigemina* negatively affects the abundance or natural distribution of *H. punctatum*. Third, that because the beetle prefers open, sunny areas (*Fields, Arnason & Philogène, 1990*; *Van Driesche et al., 2010*), the native *Hypericum* might face greater herbivory pressure in those locations and therefore be relegated to shaded areas. This phenomenon has been observed in California, where the beetle's preference for sunny environments resulted in a 99% reduction of the exotic *H. perforatum* in open areas, but not in shaded areas (*Van Driesche et al., 2010*).

To test these hypotheses, we combined field surveys with experimental manipulations, including a feeding assay testing beetle preference for the two species of *Hypericum*, a survey of the natural distribution of *Hypericum*, an experiment testing the effect of distance from forest edge on plant size and beetle damage, and a $2 \times 2$ factorial field experiment testing the effects of sun and shade on beetle damage. Specifically, we asked (1) do beetles preferentially exploit the native or exotic *Hypericum* species, (2) is beetle herbivory concentrated in sunny areas, and (3) does beetle herbivory limit the distribution of *H. punctatum* and *H. perforatum*?

## METHODS

### Study organisms

The genus *Hypericum* contains nearly 500 species spread throughout temperate parts of the world (*Nürk & Blattner, 2010*). This study focused on two species that co-occur in old-fields in central New York: the invasive *H. perforatum* and its native congener, *H. punctatum*. Both are perennials. *H. perforatum* is native to Europe, western Asia, and northern Africa, but arrived in the United States no later than the 17th century (*Josselyn, 1672*). One well-studied defensive chemical in *Hypericum* species is hypericin (*Avato & Guglielmi, 2004*; *Smelcerovic et al., 2007*), a compound that has increased toxicity in the presence of visible light (i.e., phototoxicity), with an action spectrum peak between 540 and 610 nm (*Fields, Arnason & Philogène, 1990*). In addition to its effect on invertebrate herbivores, hypericin is also toxic for grazing livestock (*Bourke, 2000*), which prompted the introduction of *Chrysolina quadrigemina* as a biological control agent for *H. perforatum* in the 1940s

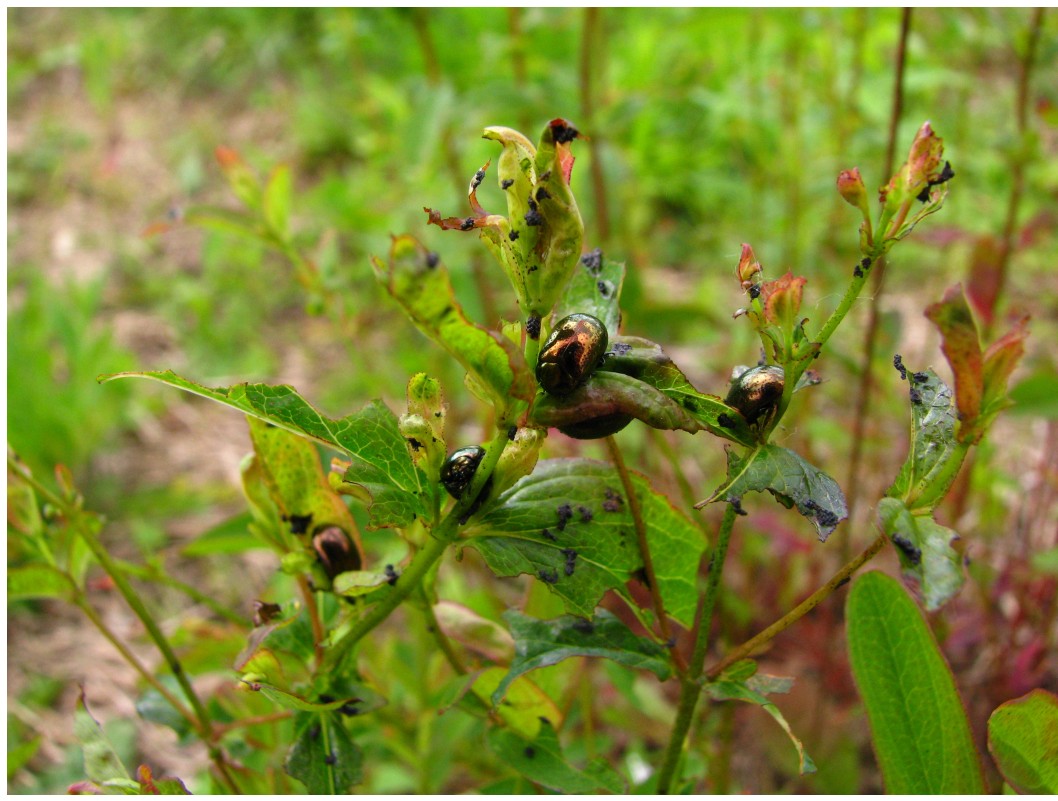

**Figure 1** *Chyrsolina quadrigemina* feeding on native *Hypericum punctatum* (photo: SCP).

(*Andres, 1985*). The native range of *C. quadrigemina* overlaps with that of *H. perforatum* in the Mediterranean region, where it consumes only *H. perforatum* and *H. tomentosum* and oviposits on *H. perforatum* (*Wilson, 1943*). Despite ingestion of phototoxic hypericin, adult *Chrysolina quadrigemina* expose themselves to sunlight by perching high up on *Hypericum* stems during the day (*Fields, Arnason & Philogène, 1990*). In laboratory experiments, however, the beetles have demonstrated negative phototropism (*Wilson, 1943*).

## Location

We conducted our field experiments in former agricultural fields belonging to Cornell University, Ithaca NY. To determine the natural distribution of *Hypericum*, we collected data in 5 fields (Durland Bird Sanctuary, 42°26′19″N, 76°23′54″W; Freese Road, 42°28′2″N, 76°26′38″W; Neimi Road West, 42°30′1″N, 76°26′7″W; Neimi Road East, 42°30′1″N, 76°26′3″W; and Whipple North, 42°29′32.13″N, 76°25′40.88″W). We established the transect experiment and sun/shade experiment in fields surrounding an experimental field station (Dryden NY, 42°27′51″N, 76°26′40″W).

## Feeding assays

To determine whether beetles preferred the native or exotic *Hypericum* species, we conducted feeding assays. In July 2011, we collected adult beetles from wild *Hypericum* plants, stored them in containers with a damp paper towel, and starved them for 24 h prior

to feeding trials. We tracked whether the beetles were collected on *H. punctatum* ($N = 17$) or *H. perforatum* ($N = 8$). We collected as many beetles as possible for the feeding trials before they began their late-summer period of inactivity, and although we attempted to collect equal numbers of beetles from each plant species on three days over the course of one week, we did not manage to find as many beetles on *H. perforatum* during the second and third days of collection. For each trial ($N = 25$), we paired undamaged stems of equal length from greenhouse grown *Hypericum* plants, placing them in a water tube and inserting these into a soil-filled pot, 10 cm in diameter. We placed one beetle on the soil surface between the two stems, encased the pots in mesh bags for 2.5 h, and recorded which plant, if any, the beetle chose to eat by the end of the trial. All trials occurred within an outdoor shade tent. To determine whether the beetles preferred the native or exotic *Hypericum*, we analyzed feeding preference with a $\chi$-square test. We excluded trials in which the beetles made no choice ($N = 11$), using only data in which the beetle fed on one of the two plants offered ($N = 14$). Beetles never fed on both species during a trial. We analyzed these data using the Wilson score interval because when sample sizes are small, the commonly used Wald interval performs poorly and the Clopper–Pearson interval gives an actual coverage probability much greater than the nominal confidence level (*Agresti & Coull, 1998*).

## Natural surveys

We assessed the natural herbivory patterns and habitat distribution of the two *Hypericum* species by recording their occurrence along transects ($N = 8$) that began at the forest edge and continued 100 m into the field, or for fields less than 200 m wide, to the midpoint of the field (two transects measured 60 m and one 40 m). To avoid biasing our samples by light exposure, we collected data from transects with differing exposures (i.e., two northeast-facing fields, two southeast-facing fields, one south-facing field, and three west-facing fields). We measured any *Hypericum* that fell within 1 m of either side of the transect, recording its distance from the forest edge and total number of leaves (as a proxy for plant size). In addition, we quantified how many leaves were heavily damaged (>50% surface area) by herbivores. Leaves with less than 50% damage usually had slime trails indicative of slug herbivory (J Tingle, pers. obs., 2010) and we excluded them from analyses of beetle damage.

Because *H. punctatum* did not occur past 40 m from the field edge (see Results), we restricted our analyses of both species to the first 40 m. We used mixed effects binomial logistic regression in R (*glmer*; package *lme4*; *Bates, Maechler & Bolker, 2012*) to examine how the number of damaged versus undamaged leaves changed with distance from the forest edge. The model included distance from forest edge and species plus their interaction as predictors, and transect as a random intercept effect. Here and elsewhere, we calculated the variance of the mixed effect model with and without the random effect (package *MuMIn*; *Bartón, 2015*) as recommended by *Nakagawa & Schielzeth (2013)*. We also examined how leaf number, our proxy for plant size, changed with distance from the forest edge, using the same predictors. Because *H. perforatum* and *H. punctatum* have very different leaf sizes, we first standardized total leaf number (i.e., total number of consumed and unconsumed

leaves) by mean leaf number per species (i.e., $x_i/\mu_x$). This eliminated the main effect of species, but enabled us to better examine the interaction between species and distance.

## Transect experiment

Given the beetle's documented preference for open, sunny areas (*Fields, Arnason & Philogène, 1990*), we established an experiment to test whether distance from nearby forest edges affected the amount of beetle herbivory that the native and exotic *Hypericum* received.

In March 2011, we began cold stratification (4 °C, 3 weeks) of *Hypericum* seeds (Companion Plants, Inc. Athens OH). We sowed the seeds in commercial potting soil (Pro-mix "BX" with biofungicide, Premier) in 36-well trays in a greenhouse, thinning them to 1 individual per cell as needed. Plants received water *ad libitum* and weekly fertilizer (21-5-20 NPK, 150 ppm). We planted the *Hypericum* into the field on 7–9 June 2011 after acclimating them in an outdoor shade tent.

We planted both species of *Hypericum* along transects in a single northeast-facing forest edge (42°28′2″N, 76°26′38″W) where both species naturally occur. Each transect ($N = 5$ pairs) began at the forest edge (0 m) and ran 50 m into the field, with a *Hypericum* planted every 5 m ($N = 11$/transect). Transect pairs consisted of a native-only and an exotic-only transect separated by 1 m, and were approximately 20 m away from neighboring pairs.

In late July 2011, we assessed herbivore damage (number of leaves heavily damaged versus undamaged). In September 2011 we harvested aboveground tissue of each *Hypericum* and of all the other plants in a 0.25 m$^2$ plot around the *Hypericum* plant, dried the material in a drying oven (50 °C), and separately determined the mass of the *Hypericum* and surrounding vegetation to 0.1 g.

To investigate how distance from edge and species identity impacted herbivory in the transect experiment, we again used mixed effects binomial logistic regression. The model included distance from edge and species plus their interaction as predictors, with transect pair as a random intersect effect. We also looked at how naturally-occurring competitor biomass changed with distance from the edge using a mixed effects linear model (*lmer* in R), with distance as the main effect and transect pair as a random spatial block. Finally, we looked at how naturally-occurring competitor biomass predicted *Hypericum* biomass with a mixed effect linear model and transect pair again as a random effect.

## Sun/shade experiment

To directly assess how herbivory impacted the growth of the native *H. punctatum*, we conducted a $2 \times 2$ full-factorial experiment that manipulated shade and protection from herbivores. We positioned 11 replicate blocks along forest edges. To prevent biases, we placed the replicate blocks along forest edges that differed in cardinal directions and ensured that each was separated from its nearest neighbor by at least 20 m. Each replicate block contained four 1 m$^2$ plots representing each of the four shade and caging combinations. We manipulated shade by locating half of the plots under the canopy at the field edge. We located paired sun plots 15 m away from the forest edge. Light measurements taken between 10:45 am and 12:15 pm on 26 July 2010 (Li-Cor, Model LI 250 Light Meter) showed that
on average, shaded plots received 30% the amount of mid-day light that sunny plots did ($t_{1,10} = -7.56$, $p < 0.0001$). Within each shade or sun manipulation, we also protected half of the plants from herbivores by completely enclosing the cages with bridal veil. The unprotected cages had bridal veil on the top only. Protected plants received heavy damage (>50% of leaf area eaten) on 8% of their leaves, versus 16% of leaves in unprotected plants ($F_{1,76} = 2.6$, $p = 0.111$). Thus, the herbivory manipulation did not completely exclude herbivores, but was effective at reducing herbivory. We planted *H. punctatum* on 21 June 2010 into each plot using the same propagation methods as those described in the transect experiment, except that we used locally collected *H. punctatum* seeds. We trimmed the resident vegetation in each plot to the ground and planted two small *H. punctatum* plants in opposite corners of the plot, approximately 1 m apart. Afterwards, we did not further manipulate the vegetation. We assessed herbivory and measured plant height on 28 June and three weeks later on 19 July 2010.

To examine how sun versus shade impacted patterns of herbivory, we used binomial logistic regression to assess the number of damaged versus undamaged leaves in the uncaged plots. To assess how sun and herbivore exposure impacted *Hypericum* growth rates, we used a factorial ANOVA with light (sun vs. shade) and protection (caged vs. uncaged) as main effects, plus their interaction, as well as replicate block as a random intersect. We calculated relative growth rate as:

$$\frac{\Delta height}{height_{initial}} \times \frac{1}{days}$$

We then ln + 1 transformed the data to improve the normality of the residuals. If both of the original *H. punctatum* individuals survived, we took the average relative growth rate of the two. One uncaged, sunny plot and one caged, sunny plot experienced complete mortality and were excluded from the analysis.

## RESULTS

### Does the beetle preferentially exploit the native or exotic *Hypericum* species?

In the feeding assays, 11 beetles chose *H. punctatum*, three chose *H. perforatum*, and 11 made no choice. A Wilson score interval at 95% confidence for the 14 beetles that made a choice does not include 0.5, indicating beetle preference for *H. punctatum* (0.52–0.92), but the small sample size leaves us little power to determine the strength of this preference.

Amongst naturally-occurring *Hypericum*, we found that natives on average had heavy damage on 23% of leaves compared to only 13% of leaves on the exotic (Fig. 2A) and this trend was consistent with distance from the forest edge (Fig. 3A). In contrast, native *H. punctatum* planted along an experimental transect received less damage than *H. perforatum* (21% of leaves *vs.* 27%; Fig. 2B). However, distance from the forest edge strongly influenced how much damage each species received (Fig. 3B; species × distance: $\chi^2 = 29.2$, $p < 0.0001$), with the exotic receiving more damage towards the forest edge and the native receiving more damage away from the forest edge.

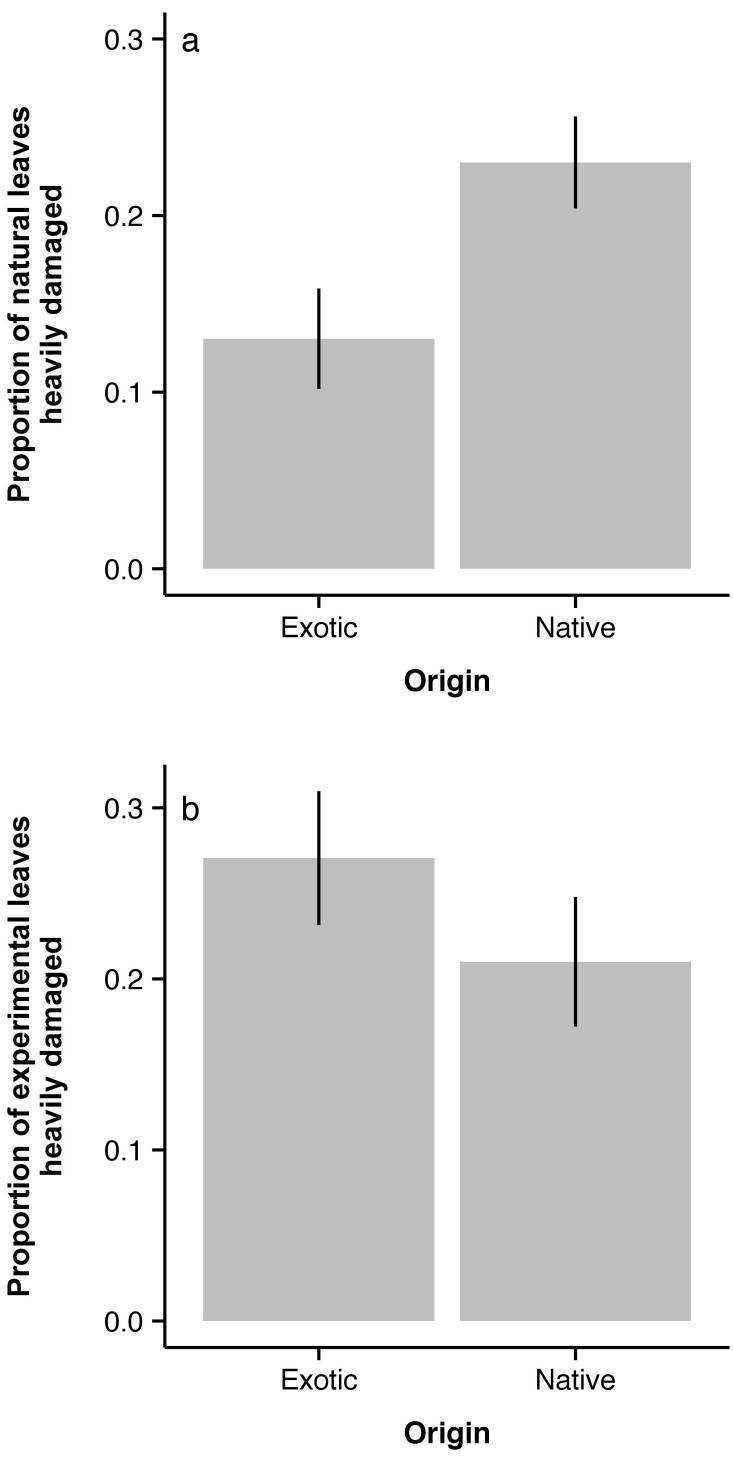

**Figure 2  Comparison of beetle interactions with native and exotic *Hypericum*.** (A) proportion of leaves heavily damaged by beetles in naturally occurring plants; (B) proportion of leaves heavily damaged in transect experiment plants.

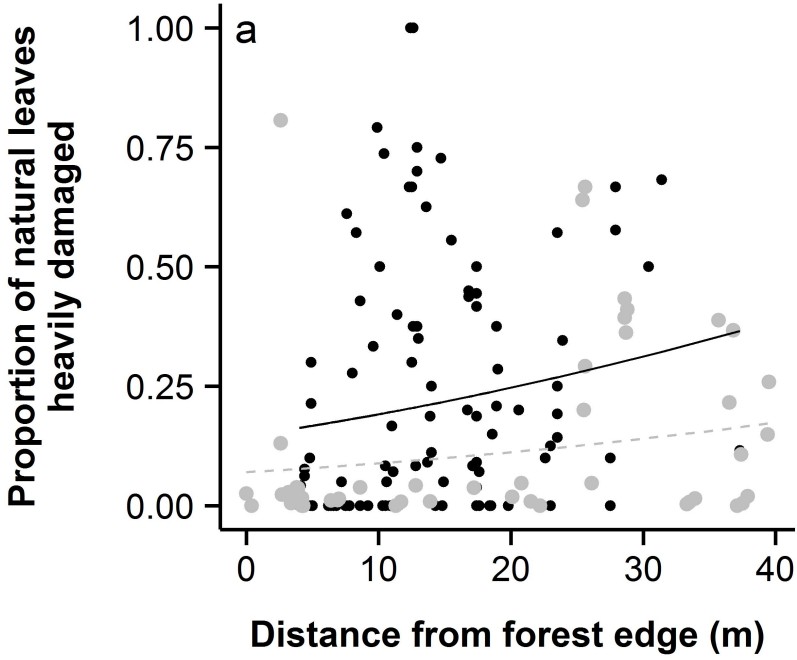

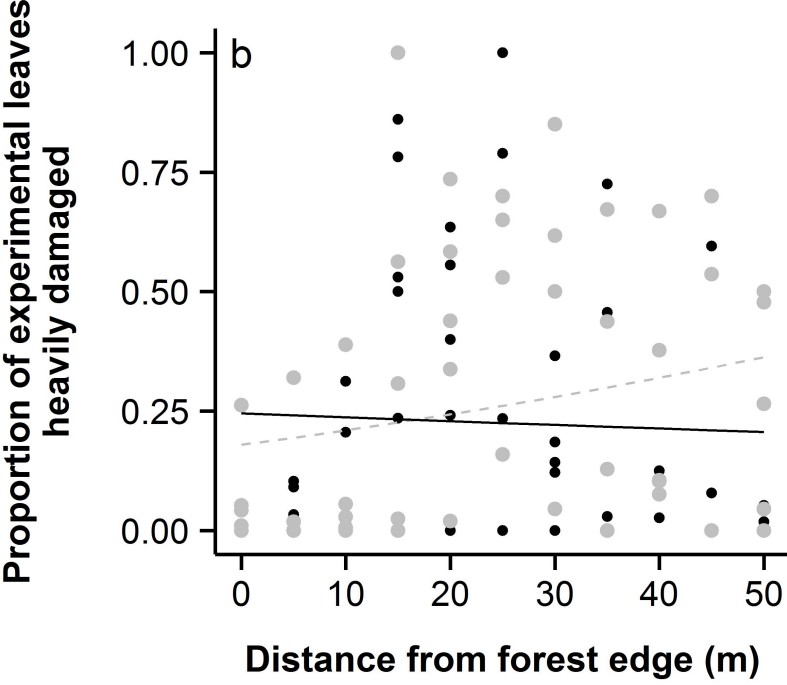

**Figure 3** **Distance from forest edge altered patterns of beetle damage in both (A) natural populations and (B) the transect experiment.** Gray points indicate *H. perforatum*, with a dashed gray line indicating the fit between the proportion of leaves heavily damaged and distance from the forest edge. Black points indicate *H. punctatum*, with a solid black line indicating the fit between the proportion of leaves heavily damaged and distance from the forest edge.

### Is beetle herbivory concentrated in sunny areas?

Distance from forest edge shaped how much heavy damage each species received. In natural populations, the likelihood of heavy damage increased with distance from forest edge for both species, but more dramatically so for *H. punctatum* than for *H. perforatum* (Fig. 3A; species × distance: $\chi^2 = 21.3$, $p < 0.0001$; $R^2_{marginal} = 0.08$; $R^2_{conditional} = 0.08$). In the transect experiment, the likelihood of damage decreased with distance from forest edge for *H. punctatum*, but increased for *H. perforatum* (Fig. 3B; species × distance: $\chi^2 = 29.2$, $p < 0.0001$). However, the random transect effect explained more of the variation in damage ($R^2_{marginal} = 0.01$; $R^2_{conditional} = 0.19$). In the sun/shade experiment, *H. punctatum* in sunny and shady uncaged plots did not differ in the amount of damage they received ($\chi^2 = 1.1$, $p = 0.300$), though small sample size may have hindered our ability to detect differences between treatments. Thus, we have limited evidence to suggest that *C. quadrigemina* feeding in our experimental fields was restricted to sunny areas.

### Does beetle herbivory limit the natural distribution of *H. punctatum*?

Native *H. punctatum* grew closer to the forest edge than did the exotic *H. perforatum* ($14.8 \pm 1.6$ m *vs.* $34.4 \pm 5.9$ m; mean $\pm$ 95% CI), and we found no *H. punctatum* past 40 m from the edge (Fig. 4). Yet, both species appeared to grow more vigorously in open, sunny areas. Although the size of *Hypericum* planted in the transect experiment did not differ with distance from the forest edge (distance: $\chi^2 = 1.4$, $p = 0.224$; species × distance: $\chi^2 = 0.2$, $p = 0.640$), *H. punctatum* in the sun/shade experiment had 48% higher growth rate in the sun than in the shade ($\chi^2 = 9.4$, $p = 0.002$) and naturally occurring individuals of both *Hypericum* species tended to be larger towards the center of the fields (distance: $\chi^2 = 5.4$, $p = 0.019$; species × distance: $\chi^2 = 2.9$, $p = 0.085$). Thus, at least two lines of evidence suggest that *H. punctatum* grew larger or more quickly in open conditions compared to the shaded conditions where it naturally occurred.

Despite the frequency of attack by *C. quadrigemina* observed across experiments, we found no evidence that beetle herbivory limited *Hypericum* growth. Excluding beetles in the factorial manipulation of shade and herbivory had no effect on relative growth rate ($\chi^2 = 0.2$, $p = 0.632$). There was also no interaction between beetle exclusion and sun/shade treatment ($\chi^2 < 0.1$, $p = 0.786$). We also found no evidence that the competitive environment limited *Hypericum* distribution. The biomass of naturally-occurring competitors in the transect experiment did not change with distance ($\chi^2 < 0.1$, $p = 0.782$) and there was no relationship between competitor biomass and *Hypericum* size ($\chi^2 = 0.9$, $p = 0.332$).

## DISCUSSION

Specialist natural enemies are often introduced as biological control agents to combat invasive species (*Simberloff & Stiling, 1996*). We found that a beetle, *Chrysolina quadrigemina*, released in North America as a biocontrol agent to control invasive St. Johnswort, *Hypericum perforatum*, readily consumed and potentially preferred a native congener, *H. punctatum*. In addition, naturally-occurring *H. punctatum* grew frequently in habitat near the forest edge, even though it usually performed better in the sun than in the

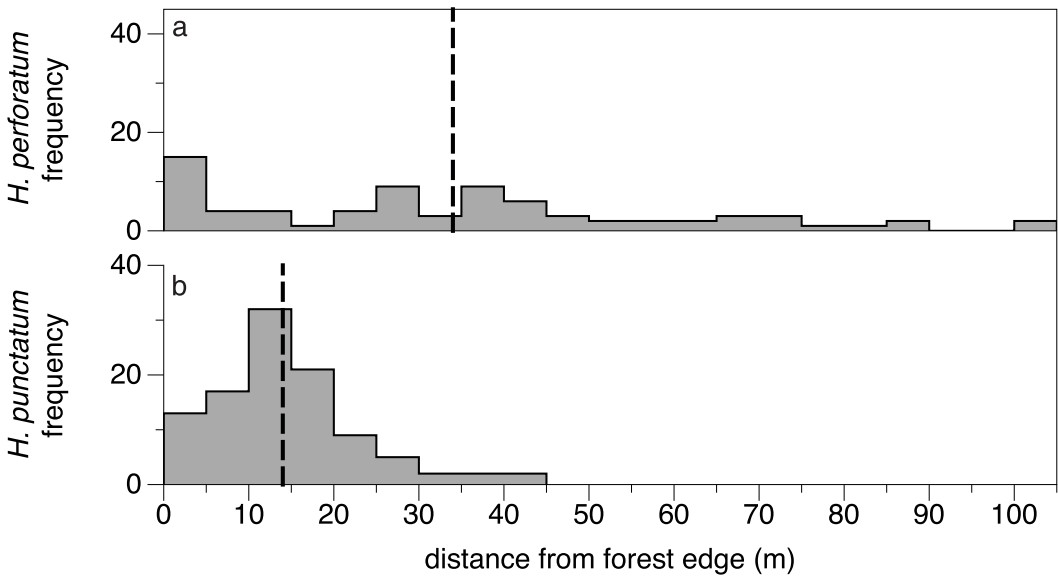

**Figure 4** Natural distribution of exotic *H. perforatum* (A) and native *H. punctatum* (B) in the field. The former occurs throughout fields, while the latter is concentrated in shadier areas less than 20 m from the forest edge and does not occur more than 40 m from the forest edge. Dashed lines indicate the mean distance from forest edge for each species.

shade, leading us to hypothesize that the beetle might affect plant distribution. However, during our short term experiments we found limited evidence to suggest that the beetle directly limited the range of *H. punctatum*.

The feeding assays and natural surveys suggest that *C. quadrigemina* beetles might prefer the native *H. punctatum* over the exotic *H. perforatum*, whereas the transect experiment showed the reverse pattern. However, all three tests show that *C. quadrigemina* readily spills over onto *H. punctatum*. Closely related plants often share similar traits that mediate interactions with insects (*Futuyma & Agrawal, 2009*), thus it is quite possible for specialist herbivores to consume species with which they share limited evolutionary history (*Agrawal & Kotanen, 2003*). Hypericin, pseudohypericin, hyperforin, and other defensive chemicals are commonly found in *Hypericum* species around the world (*Avato & Guglielmi, 2004*; *Smelcerovic et al., 2007*). The close phylogenetic relationship between native and exotic *Hypericum* likely allows *C. quadrigemina* to recognize the native as a suitable host despite sharing little evolutionary history with it. *C. quadrigemina* has displayed host-switching in the past: a population of *C. quadrigemina* in California adapted to better use the ornamental *H. calycinum* as a host, showing higher larval performance when reared on that plant than on *H. perforatum*, in contrast to beetles from other populations (*Andres, 1985*). According to the USDA Plants Database, there are more than 50 native species of *Hypericum* in North America, most of which are likely to overlap with the introduced range of *H. perforatum* and *C. quadrigemina*, given that the latter two species range across the continent (http://plants.usda.gov/core/profile?symbol=HYPER). Native *Hypericum* may also possess traits shown to drive insect preference in other systems, such as superior nitrogen content, carbon to nitrogen ratio, leaf area, etc. (*Pérez-Harguindeguy et al., 2003*).

Moreover, since North American *Hypericum species* have not coevolved with the beetle, they may lack mechanisms to resist beetle herbivory and therefore present an easier target, just as other native plant species such as *Viburnum* sp. have lacked adequate defenses against introduced herbivores (*Desurmont et al., 2011*) and some exotic plants lack resistance to native herbivores when they invade novel habitats (*Verhoeven et al., 2008*). The native plants could also suffer a disproportionate effect of leaf herbivory on their flower production or seed set.

This spillover of a biological control agent onto a congener of its target plant is not unprecedented. In one well-known case of biological control gone wrong, the weevil *Rhinocyllus conicus* fed on more than 20 native North American thistle species in the genus *Cirsium* after its introduction to control exotic thistle species, greatly reducing seed production in one native species (*C. canescens*), whose population density subsequently suffered (*Louda et al., 2003*). However, a recent review on non-target impacts of biological control agents found that >99% of the 512 agents reviewed had no known appreciable impact on native species, excepting the weevil that attacked native *Cirsium* thistles and the moth that attacked native *Opuntia* cacti, neither of which would be chosen as biocontrol agents under today's stricter risk assessment protocol (*Suckling & Sforza, 2014*). This review relied on information already existing in the prior literature, so it is quite possible that there are other unreported cases, such as that of *Hypericum punctatum*. Moreover, ecological impacts of biological control agents can indirectly impact species at multiple trophic levels. For example, biocontrol agents that share natural enemies (e.g., predators or parasitoids) with native species may indirectly suppress populations of those natives by increasing the abundance of their natural enemies (*Carvalheiro et al., 2008*). Additionally, a biocontrol agent could alter ecosystem function by providing a novel food resource for native organisms at higher trophic levels or by acting as an ecosystem engineer (*Seymour & Veldtman, 2010*). These indirect effects may be particularly difficult or impossible to predict due to ecosystem complexity. Given that these unintended effects may not appear until long after the initial release and that host choice can rapidly evolve in herbivorous insects, there is a clear need for long-term monitoring of biological control systems (*Simberloff & Stiling, 1996*; *Van Klinken & Edwards, 2002*; *Singer et al., 2008*).

Current protocol for biological control agents requires rigorous testing before the United States Department of Agriculture's Animal and Plant Health Inspection Service grants the required permits for importation and release (*USDA, 2000*). Careful laboratory testing of potential agents through a combination of feeding assays and oviposition tests is believed to reliably predict any non-target plants at risk for spillover (*Paynter et al., 2015*). In fact, our agent of interest, *Chrysolina quadrigemina*, was shown to successfully complete its life cycle on two native *Hypericum* species (*H. frondosum* and *H. moserianum*) and one exotic species used for ornamental purposes (*H. calycinum*) prior to its release in western North America, and post-release laboratory tests showed that it completed development on two more native species, *H. anagalloides* and *H. scouleri* (*Hinz et al., 2014*). Some authors argue that current protocols focus too heavily on risks rather than on benefits, such that a success story like *C. quadrigemina* controlling the toxic *Hypericum perforatum* would not be possible today due to perceived risks, even if the ecological benefits of controlling

the invasive plant outweigh the negative impacts of control agent spillover (*Hinz et al., 2014*). Thus, decisions to release a biocontrol agent must factor in likelihood of spillover, tolerance of risk, and the negative consequences of inaction.

Previous studies have suggested that *Chrysolina* beetles seek out sunny areas (*Fields, Arnason & Philogène, 1990*). This led us to hypothesize that *Hypericum* might find "enemy free space" in shady areas (*Jeffries & Lawton, 1984*), as has been seen in other systems. *Louda & Rodman (1996)* showed, for example, that insect herbivory rather than physiological adaptation restricted bittercress, *Cardamine cordifolia,* to shaded habitat. Indeed, naturally-occurring *H. punctatum* grew three times closer to the forest edge than did *H. perforatum* and were in the shade for part or all of the day. Our data suggest that forest edges represent marginal habitat, as the size of both *Hypericum* species increased with distance from forest edge in the natural survey data, and *H. punctatum* showed higher growth rates in sunny plots than in shady plots in the sun/shade experiment. Our observation that *H. punctatum* was more common on the forest edge despite its higher performance in sunnier conditions suggests that some biotic or abiotic factors limit their growth or dispersal in open fields. Moreover, we found some evidence that herbivory was higher towards the center of fields (Fig. 3), but when we reduced herbivory with insect netting in both sunny plots and shady plots, we did not observe any increase in plant growth. It is important to note, however, that our herbivore exclusion treatment halved but did not eliminate damage. It is possible that small amounts of herbivory (on 8% of the leaves) could limit growth to the same degree as herbivory on 16% of the leaves. Thus, we found no direct evidence that beetle herbivory limits *H. punctatum growth* or distribution, but can not definitively rule out the possibility.

We also have little evidence to show that competition with neighboring plants restricts the native *H. punctatum*. If plants were restricted to shaded areas by competitors, then we might expect larger competitors in sunny areas farther away from the forest edge, but there was no relationship between distance and the mass of plants surrounding *H. punctatum* in the transect experiment. However, aboveground plant mass represents only one facet of competition. Belowground plant mass, nutrient use, or chemical interactions underground could tell a different story about competition near the forest edge versus in open fields. Competitive exclusion can influence distribution of organisms (*Grime, 1973*; *Armstrong & McGehee, 1980*), as can other factors such as browsing by vertebrate herbivores, nutrient availability, soil type and depth, and water availability (*Maron & Crone, 2006*; *Chen et al., 2015*; *Martorell et al., 2015*).

In conclusion, we did not find support for the hypothesis that introduced beetles limit the habitat distribution of native *H. punctatum*. We did, however, discover that beetles significantly damage the native plant and in some cases prefer it to the exotic *H. perforatum*. Thus, we present the first documentation that *C. quadrigemina* readily spills over onto the native *H. punctatum*. While the consequences of this spillover need further elucidation, our results highlight the importance of considering congeneric natives when selecting biocontrol agents.

## ACKNOWLEDGEMENTS

We thank J Simonis, B Dalziel, and the Agrawal lab for insightful discussion and critique, and PeerJ reviewers for their constructive comments on the manuscript.

### Funding

This work was partially funded by a Howard Hughes Medical Institute fellowship to JT. The funders had no role in study design, data collection and analysis, decision to publish, or preparation of the manuscript.

### Grant Disclosures

The following grant information was disclosed by the authors:
Howard Hughes Medical Institute fellowship.

### Competing Interests

The authors declare there are no competing interests.

### Author Contributions

- Jessica L. Tingle conceived and designed the experiments, performed the experiments, analyzed the data, wrote the paper, reviewed drafts of the paper.
- Susan C. Cook-Patton conceived and designed the experiments, performed the experiments, analyzed the data, wrote the paper, prepared figures and/or tables, reviewed drafts of the paper.
- Anurag A. Agrawal conceived and designed the experiments, reviewed drafts of the paper.

### Data Availability

Data has been uploaded as Supplemental Information.

### Supplemental Information

Supplemental information for this article can be found online at http://dx.doi.org/10.7717/peerj.1886#supplemental-information.

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
