# Peer review of "Spillover of a biological control agent (Chrysolina quadrigemina) onto native St. Johnswort (Hypericum punctatum)"

_PeerJ, doi:10.7717/peerj.1886_

## Round 0.1 · original submission · Minor Revisions

This paper has particular value in addressing non-target impacts associated with biological control. Please respond to the comments and suggested revisions by the two reviewers, who were both very positive about your paper, but had some good suggestions/questions of clarity.

Reviewer 1 ·

Basic reporting

This study examined whether a biocontrol agent had switched to a congeneric of the invasive species it was introduced to control. The study contains both surveys and experimental aspects, which I liked. I also liked that the study didn’t just assess whether the biocontrol is attacking the native species but also assessed whether the biocontrol was having an effect on the native species’ distribution.
The paper is well communicated for the most part and there was little that I queried from an understanding point of view.

Experimental design

I think that some of the experiments suffered from low sample size, which would have affected their ability to detect significant effects. But otherwise, I have no issues with things like pseudoreplication.

Validity of the findings

I also thought that the models should have been more completely explained (for example, we are not informed of how much variance the models explain). The fits didn't look great, so I think that sort of information should be provided.

Additional comments

I thought that the study had a slightly simplistic approach to biocontrol and that perhaps this would be remedied with just a few lines about indirect effects that biocontrol can have – e.g. apparent competition (see Carvalheiro et al. 2008) or the creation of resources by biological control agents (see Seymour & Veldtman 2010), as ecology is far more complicated than simply one species interacting with another, other species are affected. As ecologists, we need to consider the entire context. But otherwise, thank you, I enjoyed this paper.
Specific comments
Abstract
Needs to mention where this study takes place!
Methods

Line 92 – the sample size (n=14) strikes me as a little small – see if this reflects in the results…
Line 111 – I take it you mean random effect by “random spatial block”. Is this a random slope, random intercept or random slope and intercept model that you settled on in the end?
Lines 149-151 – I’m sure this is easily explained but I couldn’t understand how you had five different forest edges but then you list 11 different blocks. Could you just clarify this?
Line 173 – the three level fraction makes no sense to me, would it be possible to rather use mathematical notation to communicate how you calculated this?
Results
Line 182 – insert comma before “fig 2a”
Lines 184-187 – this information about Wilson scores and performance of Clopper Pearson etc. should rather appear in the methods section.
Lines 188-191 – your methods say that you performed mixed models on your transect data but then you do not report on the significance of these relationships? I cannot tell if 21% damage is significantly lower than 27% damage, made more difficult by the fact that standard deviations are not reported. The models used to assess damage by species and at different distances to forest edge SHOULD tell you if one species received significantly more damage than the other because species is one of your explanatory variables, and you account for distance in the model.
Line 190 – when you speak of transect experiment, is this the survey you did, and not an actual experiment?
Line 201 – Is it possible that you do not detect an effect in the sun/shade experiment owing to small sample size?
Line 232 –beetle damage to leaves might mean a larger effect on flower production or seed set of one Hypericum over the other? So a more indirect effect of herbivory is also possible?
Line 287 – I think also some mention of other issues – i.e. not direct transfer from one host to another, but indirect interactions are often ignored or not thought of. Ecology is complicated and perturbations seldom affect only one (target) species. I think this is an important point to make – see, e.g. about apparent competition (see e.g. Carvalheiro et al. 2008) and about creating resources for species that ordinarily would not have had these (see, e.g. Seymour and Veldtman 2010), and thus changing the invertebrate community overall:
Carvalheiro, L. G., Y. M. Buckley, R. Ventim, S. V Fowler, and J. Memmott. 2008. Apparent competition can compromise the safety of highly specific biocontrol agents. Ecology letters 11:690–700.
Seymour, C. L., and R. Veldtman. 2010. Ecological role of control agent, and not just host-specificity, determine risks of biological control. Austral Ecology 35:704–711.
Figures
I thought figure 2a was perhaps not necessary, given that this is reported in the text already. Also, the standard deviations on the figures appeared to be the same size as each other, in both figs 2b and 2c, is this correct?
Figure 3b – the data look rather messy relative to the fitted line, can you tell us how much variation is explained by these models?

·

Basic reporting

Basic reporting is more than sufficient, and I only have a few minor comments:

In line 28 please state where has the exotic weed been successfully controlled.

Lines 89 and 126 - what is the percentage shading of the shade tent?

Line 260 - please give the Genus name of the thistle species.

Experimental design

The manuscript has several experiments that were conducted. I do not find any flaws in any of the experiments done.

Just two comments:

Lines 84-85, were beetles collected from either the native or the exotic equal in number and were individuals for each trial randomly chosen? Please elaborate.

In line 181, why were only 25 assays done and not 50, to strengthen the power of the said analysis?

Validity of the findings

I find the findings of the manuscript to be valid.

Only two comments:

Lines 300-302, experiment to exclude herbivores only reduced herbivory and did not completely exclude it. This could have resulted in a lack of a difference in plant growth between exclusion and control. Please consider this in discussion.

If native congeners of target weeds are "considered" and found to be either eaten or oviposited on by the biological control agent, why are there still cases where the agent is released? How is the expected impact on the target weed gauged prior to release?

Additional comments

Nice study that has entailed much work. Good to see more information on the contentious issue of non-target impacts.

---

## Round 0.2 · accepted · Accept

I am satisfied with your responses to the queries raised by the reviewers, thank you.